# EEG/fNIRS Based Workload Classification Using Functional Brain Connectivity and Machine Learning

**DOI:** 10.3390/s22197623

**Published:** 2022-10-08

**Authors:** Jun Cao, Enara Martin Garro, Yifan Zhao

**Affiliations:** School of Aerospace, Transport and Manufacturing, Cranfield University, Bedfordshire MK43 0AL, UK

**Keywords:** sensor fusion, mental workload, n-back, artificial intelligence, feature engineering

## Abstract

There is high demand for techniques to estimate human mental workload during some activities for productivity enhancement or accident prevention. Most studies focus on a single physiological sensing modality and use univariate methods to analyse multi-channel electroencephalography (EEG) data. This paper proposes a new framework that relies on the features of hybrid EEG–functional near-infrared spectroscopy (EEG–fNIRS), supported by machine-learning features to deal with multi-level mental workload classification. Furthermore, instead of the well-used univariate power spectral density (PSD) for EEG recording, we propose using bivariate functional brain connectivity (FBC) features in the time and frequency domains of three bands: delta (0.5–4 Hz), theta (4–7 Hz) and alpha (8–15 Hz). With the assistance of the fNIRS oxyhemoglobin and deoxyhemoglobin (HbO and HbR) indicators, the FBC technique significantly improved classification performance at a 77% accuracy for 0-back vs. 2-back and 83% for 0-back vs. 3-back using a public dataset. Moreover, topographic and heat-map visualisation indicated that the distinguishing regions for EEG and fNIRS showed a difference among the 0-back, 2-back and 3-back test results. It was determined that the best region to assist the discrimination of the mental workload for EEG and fNIRS is different. Specifically, the posterior area performed the best for the posterior midline occipital (POz) EEG in the alpha band and fNIRS had superiority in the right frontal region (AF8).

## 1. Introduction

Mental workload refers to the amount of working memory required to complete a task in a specified time. Its assessment has attracted many researchers, and workload has been characterised by a variety of physiological sensor data. Investigation of mental workload in neuroscience is significant for a variety of reasons. First, a person’s high cognitive workload will affect learning capacity and cause distraction [1]. Second, since there is a limit to the size of a cognitive workload, there is also a limit to an individual’s performance in a given cognitive activity [2]. As a result, assessing mental workload is important for preventing accidents in many areas [3]. Table 1 compares various popular neuroimaging modalities for evaluating mental workloads: such as functional near-infrared spectroscopy (fNIRS), electroencephalography (EEG)/Magnetoencephalography (MEG), functional magnetic resonance imaging (fMRI), and position emission tomography (PET).

Because it has the advantages of low cost and high temporal sampling rate, EEG has been well-accepted in the field of disease prediction [4], sleep stages [5], and brain stimulation for different neurological workloads [6] as well as mental workload evaluation [7]. A substantial number of studies have reported a significant EEG spectral correlation with workload in stereotypical frequency bands: such as delta (1–4 Hz), theta (4–7 Hz), alpha (8–15 Hz), and beta (16–31 Hz) [8,9,10]. Several popular machine learning methods have been applied using EEG features such as support vector machine (SVM) [11], naive bayes [3] and linear discriminant analysis (LDA) [12,13]. Although those methods achieved satisfactory mental workload classification results, it was notable that most EEG-based features were extracted from a single channel that was univariate-based and neglected association between channels. As a multivariate approach, functional brain connectivity (FBC) is statistically interdependent among spatially distant neurophysiological regions [14,15,16]. It has been proven that FBC reveals the underlying function of different brain regions and their complex cortical intercommunication, which helps improve understanding of many neurological conditions including brain-related disorders and emotions [15,16,17]. Kakkos et al. [7] fed univariate spectrum power features and FBC estimations from EEG into several machine-learning classifiers and achieved promising results in two-level workload discrimination. However, the potential of FBC in multiclass workload classification problems, particularly in combination with other sensing modalities, has not been fully explored. 

In recent decades, fNIRS has grown rapidly as a tool for monitoring functional brain activity in a wide range of applications and populations. fNIRS devices detect two hemodynamic signals, oxygenated (HbO) and deoxygenated (HbR) hemoglobin, from the cortical surface at a spatial resolution of 2–3 cm [18,19,20]. One of the main reasons for the increased interest in using fNIRS for cognitive activities is that it is resistant to motion artefacts [21], which is usually a big problem for EEG data acquisition. Furthermore, fNIRS can be more precise in brain activation areas due to its relatively high spatial resolution. As a result, fNIRS overcomes some shortcomings of EEG. The importance of including both HbO and HbR for analysis has been emphasised by a few studies because their combination provides a more comprehensive assessment of cortical activation [22,23,24,25]. The majority of related studies has focused on using mean values [26,27], standard deviation [27] and slope [22,25].

Some researchers explored the effectiveness of using both EEG and fNIRS information for n-back workload classification. Liu et al. [28] employed LDA and obtained 68.1% classification accuracy in the n-back working-memory task using a combined EEG–fNIRS approach, but it used univariate features based on a single channel. Saadati et al. [29] used deep neural networks and hybrid EEG–fNIRS features. It was claimed that the classification accuracy is considerably higher than that of EEG or fNIRS alone. However, there is very limited research on EEG brain connectivity combined with fNIRS, so the potential of using both signals to discriminate multi-level workloads requires further exploration.

In this paper, we propose a hybrid EEG–fNIRS approach to discriminate among multi-level mental workloads: univariate frequency and bivariate FBC features are extracted from EEG, and biomarkers of HbO and HbR are estimated from fNIRS. Overall, combining EEG and fNIRS tended to provide two distinct sources of information on the brain including electrical activity and hemodynamic responses; this combination has the benefits of non-invasiveness, robustness to motion, availability for all possible participants, silence, portability and cost-effectiveness. The novelty of this study is summarised in four folds:To the best of the authors’ knowledge, this study is the first to use combined features of EEG-based FBC and fNIRS for workload estimation.This paper explores different linear and nonlinear FBC representations in the time and frequency domains with their associated effect on classification accuracy.This study reports the contribution of different regions to the classification accuracy of the two sensing modalities.Topographic and heat maps were used to reveal distinct areas where the greatest change occurred at different workload levels.

## 2. Materials and Methods

As shown in Figure 1, the proposed framework contains four main steps: data preprocessing, feature extraction, feature selection, and machine-learning classification. The detail of each step is as follows:

### 2.1. Dataset

This study made use of a dataset gathered by Shin et al. [30] at the Technische Universität Berlin. The dataset comprised scalp recordings—30 EEG channels and 36 fNIRS channels—for mental workload during n-back tasks. The channels and their locations are shown in Appendix A. These activities were divided into four categories: 0-, 2-, and 3-back tasks, as well as rest between tasks. Twenty-six healthy, right-handed people took part, and the dataset was divided into three sessions, each with three randomly organized sets of 0-, 2-, and 3-back tasks, meaning that each participant completed nine sets of n-back tasks. A single task consisted of a 2 s instruction indicating the type of task (0-, 2-, or 3-back), a 40 s task period that consists of 20 trials, a 1 s stop period, and a 20 s rest period (see Figure 2). Therefore, there were 26 × 3 × 9 = 702 tasks available for all participants. All EEG and fNIRS signals were captured at the same time.

### 2.2. Signal Preprocessing and Feature Extraction

#### 2.2.1. fNIRS

fNIRS data were preprocessed using the BBCI toolbox in MATLAB R2019b [31]. The sampling rate was 10 Hz. Initially, HbR and HbO values were calculated using the modified Beer–Lambert equation (mBLL) from the fNIRS optical density [32]. A sample of HbR and HbO values for each participant is shown in Appendix A. Data augmentation was performed to create small informative segments. To reduce noise and artefacts, fNIRS signals were passed through a third-order digital Butterworth filter between 0 and 0.04 Hz. Additionally, baseline correction was applied to the fNIRS signals to remove the intra-individual variance of the starting values. In this step, the segments were normalised by subtracting the median value of the pre-stimulus baseline from the signal in each segment [8]. 

It should be noted that there was a general 6 s delay between the stimulus representation and peak cortical hemodynamics. This delay was determined by the task and HbR and HbO concentrations. Normally, the cerebral hemodynamic response does not return to baseline until 10 s after stimulus presentation. However, agreement on an ideal time window for analysis had yet to be reached because the best temporal length depended on the paradigm used and participant characteristics, such as age [21]. This paper conducted a sensitivity analysis to identify the optimal time window to produce the most accurate mental workload estimate and a size of 5s was used. The window slides through the whole 40 s period with a 1 s step. This analysis was performed independently for each participant. 

#### 2.2.2. EEG

EEG data were also preprocessed using the BBCI toolbox in MATLAB R2019b, and resampling was done at 200 Hz. The improved weight-adjusted second-order blind identification (iWASOBI) method in the automatic artifact removal (AAR) toolbox in EEGLAB was used to gain ocular artifact rejection. Initially, data augmentation was done by segmenting data samples into smaller but still informative segments. Then, the data were bandpass-filtered between 1 and 45 Hz using a third-order Butterworth digital filter. The EEG epochs were extracted from −500 to 6000 ms with respect to the onset of every stimulus. Power spectral density (PSD) was calculated for three frequency bands of EEG recordings: delta (0.5–4 Hz), theta (4–7 Hz), and alpha (8–15 Hz) since previous studies indicated that low-frequency information made more contributions for measuring mental workload [7,33]. The FBC was estimated using four methods: Pearson correlation coefficient (PCC), mutual information (MI) in the time domain, magnitude squared coherence (MSC), and phase-locking value (PLV) in the frequency domain. The principal details are given as follows:

The PCC was able to evaluate the linear interdependency between two signals in the time domain and ranged from −1 to +1. The correlation coefficient between signals x and y were
(1)$$\rho_{xy}=\frac{E\left[\left(x-\mu_{x}\right)\left(y-\mu_{y}\right)\right]}{\sigma_{x}\sigma_{y}}$$
where *E* is the expected value; $\mu_{x}$ and $\mu_{y}$ are the mean values; and $\sigma_{x}$ and $\sigma_{y}$ are the standard deviations of the x and y time series. 

MSC is a linear method to estimate interconnections between two signals in the frequency domain calculated by PSD. The MSC of signals x and y can be written as
(2)$${\mathrm{MSC}}_{xy}\left(f\right)=C_{xy}^{2}=\frac{S_{xy}{\left(f\right)}^{2}}{\left|S_{xx}\left(f\right)\right|\times \left|S_{yy}\left(f\right)\right|}$$
where $S_{xx}\left(f\right)$ and $S_{yy}(f$) are the PSDs of signals x and y, respectively; and $S_{xy}\left(f\right)$ is the cross PSD at frequency f.

According to information theory, the MI of two random variables, x and y, shows how one is informative for the other one. Let, P(x) and P(y) be the probability distributions of random variables x and y, respectively. The entropy of x and y is defined as
(3)$$H\left(x\right)=-\sum_{j=1}^{N}P\left(x_{j}\right)\log_{b}\left(P\left(x_{j}\right)\right)$$
(4)$$H\left(y\right)=-\overset{N}{\underset{j=1}{\sum }}P\left(y_{j}\right)\log_{b}\left(P\left(y_{j}\right)\right)$$
where *N* defines window length. H(y|x) and H(x,y) represent conditional entropy and joint entropy between x and y, defined respectively as
(5)$$H\left(x,y\right)=-E_{x}[E_{y}[\log_{b}P\left(x,y\right)]]$$
(6)$$H\left(y|x\right)=-E_{x}[E_{y}[\log_{b}P\left(y|x\right)]]$$
where *E* is the expected value function. The MI of two random variables x and y is computed as follows
(7)MI(x,y)=H(x)+H(y)−H(x,y)=H(y)−H(y|x)
MI(x,y)=0 if and only if random variables X and Y are statistically independent. Notably, the MI is a nonlinear method in the time domain,

Phase synchronisation (PS) assumes that two oscillation systems without amplitude synchronisation can have phase synchronisation. The phase locking value (PLV) is frequently used to obtain the phase synchronisation strength [14]. The instantaneous phase of a signal X is given by
(8)$$\emptyset_{x}\left(t\right)=arctan\frac{\tilde{x}\left(t\right)}{x\left(t\right)}$$
where $\tilde{x}\left(t\right)$ is the Hilbert transform of x(t) which is defined as
(9)$$\tilde{x}\left(t\right)=\frac{1}{\pi }PV\int_{-\infty }^{+\infty }\frac{x\left(\tau \right)}{t-\tau }d\tau$$
where PV refers to the Cauchy principal value. The PLV for two signals is defined as
(10)$$\mathrm{PLV}=\left|\frac{1}{N}\sum_{j=0}^{N-1}e^{j\left(\emptyset_{x}\left(j\Delta t\right)-\emptyset_{y}\left(j\Delta t\right)\right)}\right|$$
where Δt defines the sampling period, and N indicates the sample number of each signal [34]. The range of PLV was from 0 to 1, where 0 showed a lack of synchronisation and 1 indicated strict phase synchronisation. Notably, the PLV is a nonlinear method in the frequency domain.

### 2.3. Feature Selection and Fusion

A large number of features were extracted from EEG and fNIRS. To be more specific, considering three frequency bands (delta, theta and alpha), 28 channels and four FBC methods, there were 3 × 28 = 84 PSD features and 3 × 28 × (28−1)/2 × 4 = 4536 FBC features estimated from the EEG recording. According to the time window analysis of the fNIRS signals, the top-10 best time windows were chosen. Considering the number of channels, there were 10 × 36 = 360 features for fNIRS. The next step was to feed the extracted features into machine learning classifiers to classify three workloads/tasks. To avoid the overfitting problem of machine learning and compare the combined methods fairly with the methods using a single type of feature, a statistical significance test is used to reduce the feature number. One-way analysis of variance (ANOVA) was used to evaluate the significance of differences in the 0-back vs. 2-back vs. 3-back features. The *p*-value was the criterion for selecting the significant features. As a result, the top-10 features with the smallest *p*-values were individually selected from EEG-based and fNIRS-based techniques as classifier input. Furthermore, the top-5 features from EEG (univariate features only) and fNIRS, respectively, were combined, resulting in 10 hybrid features for comparison purposes.

### 2.4. Machine-Learning Classification

The SVM was applied to achieve workload classification. It constructed an optimal separating hyperplane in the feature space based on the structural risk minimization principle. The selected features extracted from EEG and fNIRS were fed into the SVM with a radial basis function (RBF) kernel. Different machine-learning algorithms were tested and compared, such as the k-nearest neighbour (KNN), decision tree and LDA. The SVM outperformed other methods in classification. Hence, this paper mainly used the SVM with RBF to represent classification results. To avoid overfitting =in the case of limited data, a five-fold cross-validation technique was employed. To be more specific, the dataset of each condition was divided into five subsets, and then five iterations were undertaken to ensure each subset was used for training and testing [15]. That is to say, for each iteration, 80% of the dataset was used for training and the remaining 20% for testing. Consequently, the classification result was calculated by averaging the accuracies from 5 iterations. Totally, there were 3 workload levels × 3 series × 3 sessions × 26 participants = 702 samples. Before being fed into the classifier, the features were normalised from −1 to 1 for each participant to reduce the influence of individual differences.

## 3. Results

### 3.1. Time Interval Selection

The selection of the time interval relied on the classification performance implemented on each participant. Figure 3A represents the mean classification accuracy for all participants using fNIRS-based features against the moving time window, and sustained growth was observed during the first 30 s. After a 25 s oscillation, the accuracy reached its peak when the 45–50 s time window was used. Consequently, the 10 time-windows in the range of 45 to 54 s were selected for the next step of feature extraction. Figure 3B illustrates the changes in classification accuracy against the length of the time window of fNIRS and EEG. Notably, the accuracy of using fNIRS-based features decreased along with the window-size increment for all three classification groups. However, the EEG-based method performed better following the window-size increment and peaks at 40 s, particularly for 0-back vs. 2-back and 0-back vs. 3-back. As a result, the final window-size selections for fNIRS and EEG were 5 and 40 s, respectively. Furthermore, it indicated that the fused features outperformed features from a single modality.

### 3.2. Machine-Learning Classification Performance

To select the optimal FBC features, four different methods (MI, PCC, MSC and PLV) were tested individually, and the nonlinear time-domain method, MI, was found to provide the highest classification accuracy. Figure 4 shows the comparison of the four estimations in the three bands for top-10 average classification accuracy. The error bar shows the accuracy from each iteration of cross-validation. Therefore, MI was selected as the EEG-based FBC feature for the following analysis.

To classify multi-level mental workload, the classification task was separated into three groups: 0-back vs. 2-back, 0-back vs. 3-back and 2-back vs. 3-back. The performances using EEG-based features only, fNIRS-based features only, and hybrid features were evaluated and shown in Table 2, Table 3 and Table 4. To ensure classification fairness, each classification task used 10 features as the input. The features were selected according to the significance test and a sample is given in Appendix A. The EEG alpha band information had the best performance in discriminating the three workload levels for both univariate (PSD) and bivariate features (FBC). Meanwhile, the results also suggested that the FBC features performed better with an approximately 5% accuracy increment for all three sub-tasks. When it came to fNIRS, HbR outperformed HbO, but the accuracies were both significantly lower than for EEG-based FBC features, particularly for 0-back vs. 2-back and 0-back vs. 3-back. Other references suggested that classifiers, such as LDA, SVM and CNN, achieved higher accuracy using HbR indicators [18,19]. 

Overall, the fused features (EEG–fNIRS) improved classification performance. For the 0-back vs. 2-back and 0-back vs. 3-back tasks, the hybrid method obtains the highest accuracy with 77% (Table 2) and 83% (Table 3), which suggests, as expected, there is more difference between 0-back and 3-back than between 0-back and 2-back. However, the difference between 2-back and 3-back was small, as evident by a much lower accuracy. Notably, the results suggested that the hybrid features did not have superiority in all tasks. As shown in Table 4, the FBC features in the alpha band had the best performance (62%) but the fused features had only 59%. Nevertheless, 2-back and 3-back were difficult to distinguish for any features.

To further evaluate the machine learning algorithms performance, accuracy (*Accu*), sensitivity (*Sens*) and specificity (*Spec*) were calculated:(11)$$Accu=\frac{TP+TN}{TP+TN+FP+FN}\times 100\%\;\;$$
(12)$$Sens=\frac{TP}{TP+FN}\times 100\%\;$$
(13)$$Spec=\frac{TN}{TN+FP}\times 100\%\;$$
where *TP* = True Positive; *FN* = False Negative; *TN* = True Negative; and *FP* = False Positive. Moreover, the receiver operating characteristic (ROC) curve, and the area under the ROC curve (AUC) [35,36] were used to assess the goodness of classification. Specifically, the ROC was constructed from the true positive rate (TPR = sensitivity) in the vertical axis and the false positive rate (FPR = 1-specificity) in the horizontal axis [37]. The resulting accuracy, sensitivity, specificity and AUC are shown in Table 5. The ROC curves for three binary classification tasks is shown in Figure 5.

### 3.3. Visualisation

To further explore the difference among the three workload levels, a distinct visualisation method was employed. A topographic map was used to represent the PSD distribution of the EEG alpha-band (Figure 6), which provided about 70% classification accuracy for the 0-back vs. 2-back and 0-back vs. 3-back tasks. The averaged PSD distribution across all participants, illustrated by the left column, suggested that the posterior area of 0-back had much higher PSD than 2-back and 3-back, while other areas had similar PSD distribution. It seemed that, during the low workload level, there was more brain activity in the alpha band in the posterior area than during high workload. It matched the classification result, which revealed that the posterior midline occipital (POz), left occipital (O1) and right occipital (O2) channels contributed more than the others. The patterns of 2-back and 3-back are very similar for the whole bran, which explains the low classification accuracy (55%) in Table 4. The individual PSD distribution, illustrated in the middle column of Figure 6, indicates the difficulty of the classification to an extent. Furthermore, to validate the observation, the right column of Figure 6 shows the accuracy of using each channel’s PSD as the input. As expected, the posterior area can provide more than 70% accuracy for 0-back vs. 2-back and 0-back vs. 3-back tasks. 

The topographic map of HbR features is shown in Figure 7**.** Similar to EEG, the averaged HbR distribution of 0-back is significantly different from that of 2-back and 3-back, shown in the left column. More specifically, the frontal-right area has increased HbR following the increment of workload level while the frontal-centre and middle-left areas have decreased HbR following the increment of workload level. All these findings have been supported by the classification result of individual channels (Figure 7). There is no significant difference between 2-back and 3-back in terms of the overall pattern. The individual HbR distribution is illustrated in the middle column of Figure 7. Furthermore, to validate the observation, the right column of Figure 7 shows the accuracy of using each channel’s HbR as the input. It is noted that the right frontal area can provide more than 70% accuracy for the 0-back vs. 3-back task. Interestingly, the accuracy of the posterior area (PPOz) was close to 70% for 0-back vs. 2-back and 0-back vs. 3-back tasks, which was not easy to observe from the feature visualisation. It also matched the findings in the EEG analysis.

To visualise the FBC features, a heat map was used as shown in Figure 8. The maps for individual participants are illustrated in Figure 8A–C for 0-back, 2-back and 3-back respectively. The accuracy of the three classification tasks is illustrated in Figure 8D–F for 0-back vs. 2-back, 0-back vs. 3-back, and 2-back vs. 3-back, respectively. It shows that each participant had a similar FBC pattern estimated by MI, while the value of different regions varied. Furthermore, it helped us to understand the differential contribution of the various brain regions for mental workload discrimination. The functional brain connectivity between frontal channels and Fp1 estimated by MI had a significant increase when the workload level became higher. It was also proved in Figure 8D–F that the following pairs left frontopolar–anterior midline frontal (Fp1:AFz), left frontopolar–left frontal (Fp1:F1), left frontopolar–right frontopolar (Fp1:Fp2) and left frontopolar–right frontal (Fp1:F2) provided relatively higher classification accuracy when differentiating 0-back from 2-back and 3-back.

## 4. Discussion

A comparative analysis of previous research and the proposed work employing EEG and fNIRS in mental workload classification is shown in Table 6. This paper now discusses the results in detail from three aspects: EEG vs. fNIRS, univariate vs. multivariate features, and independent vs. hybrid feature.

### 4.1. EEG vs. fNIRS

On one hand, EEG needed a longer data length to suggest difference between different-level workloads. To be more specific, a 40 s time window was the most suitable, while 5 s was suggested for fNIRS. That is to say, fNIRS required less response time to support a satisfactory classification accuracy, which meant it may be more efficient in actual application. On the other hand, the EEG-based features, especially the FBC, represented obvious advantages over the fNIRS-based features in classification accuracy although the FBC methods were more complicated and entailed a higher computational cost. 

The best region for assisting in the discrimination of the mental workload was different for EEG and fNIRS. Specifically, the posterior area performed the best for EEG (POz) in the alpha band and fNIRS had superiority in the right frontal region (AF8). Some studies suggested similar findings. Brouwer et al. [33] found the alpha power of the midline parietal (Pz) region in EEG recordings significantly decreased with memory load, effectively distinguishing 2-back from 0-back. Chu et al. [39] stated that the alpha-power of O1 indicated differences between multi-level workloads. Regarding fNIRS, the prefrontal areas were well-accepted for measuring variations in mental workload [40,41,42]. However, there was limited research pointing out a determined channel that contributes the most. Our study narrowed down the region (right frontal) to support the discrimination of workloads, as evidenced by the topographic visualisation of the machine-learning classification results.

### 4.2. Univariate vs. Multivariate Features

Considering the EEG features, the bivariate FBC approaches obtained more satisfactory accuracy compared to the univariate PSD features. The results of this study provide evidence to support the hypothesis that the FBC not only estimated the informational intercommunication of separate brain regions but also tracked distinct changes for different levels of workload. There are other supporting studies for this hypothesis in the literature on workload classification. Pei et al. [43] suggested the fusion of band power and FBC features, which were estimated by PLV and the phase lag index (PLI), enhanced the classification performance of workload identification. The PLI-based FBC was also used by Kakkos et al. [7], and the study implied that using FBC emphasised its ability to serve as a promising indicator for different workload levels. Our framework employed four FBC estimations that illustrated connections with various properties, and MI outperformed PCC, MSC and PLV for the highest classification accuracy. In this case, the proposed framework deepened the use of the FBC technique in the field of mental workload discrimination. Furthermore, it implied that, among different levels of workload, the greatest changes occurred in nonlinear brain connectivity. 

### 4.3. Independent vs. Hybrid Feature

The hybrid features of EEG and fNIRS outperformed the independent category of features in classification results, achieving the highest accuracy of 77% for 0-back against 2-back and 83% for 0-back against 3-back. It meant that different methods explored distinct information and became complementary to each other thereby improving classification performance. The results agreed with the conclusion in the literature [29,38,39,42]. The present paper is an advance on previous studies because it generated new knowledge about regional information by comparing the foci of independent types of features. To an extent, it paved the way to use an EEG–fNIRS hybrid sensor in real-world workload classifications.

## 5. Conclusions

In this paper, a novel solution relying on hybrid EEG–fNIRS features was proposed to deal with multi-level mental workload classification supported by machine-learning classifiers. To be more specific, the univariate PSD and four bivariate FBC features were extracted from an EEG recording in three frequency bands. With the assistance of HbO and HbR indicators from fNIRS, the fused features improved classification performance. Moreover, topographic and heat-map visualisation indicated distinct regions for EEG and fNIRS that represented difference among 0-back, 2-back and 3-back. Overall, the FBC technique based on an EEG recording proved its value in mental workload classification, and accuracy improvement emphasised the effectiveness of the hybrid EEG–fNIRS. The one limitation of this study was that there was a volume conduction effect in the EEG dataset, but the high classification accuracy suggested that the functional connectivity was effective for classifying different workloads. One potential future work would be to use bipolar channels rather than unipolar channels or to pre-process the data to mitigate volume conduction.

## Figures and Tables

**Figure 1 sensors-22-07623-f001:**
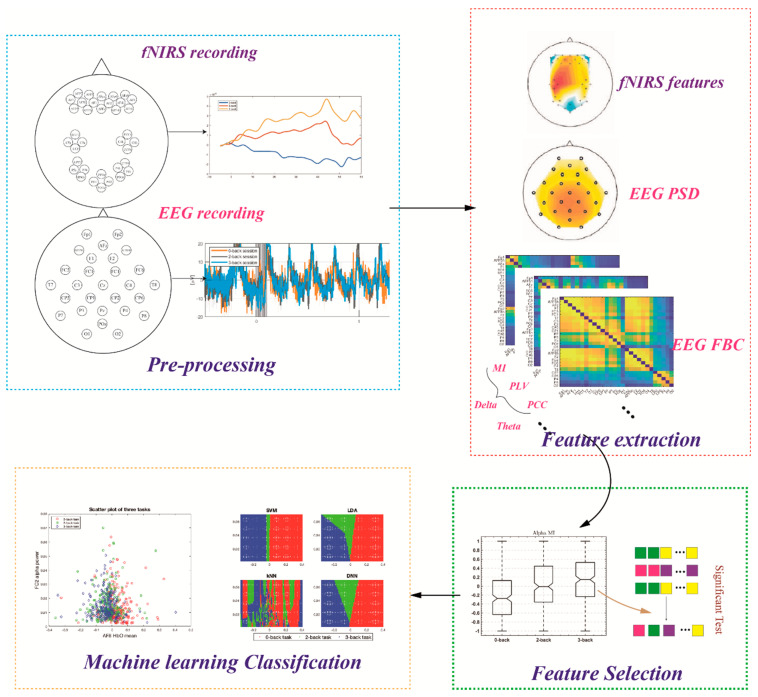
Flowchart of the proposed framework. The pipeline contains four main steps: pre-processing, feature extraction, feature selection and machine-learning classification.

**Figure 2 sensors-22-07623-f002:**
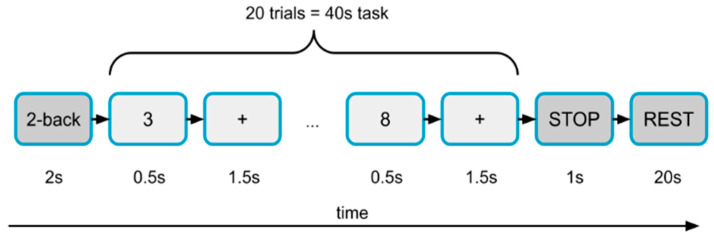
Layout of a set in the experiment. A single task consisted of a 2 s instruction indicating the type of task (0-, 2-, or 3-back), a 40 s task period that consisted of 20 trials, a 1 s stop period, and a 20 s rest period. Each participant completed nine sets of n-back tasks.

**Figure 3 sensors-22-07623-f003:**
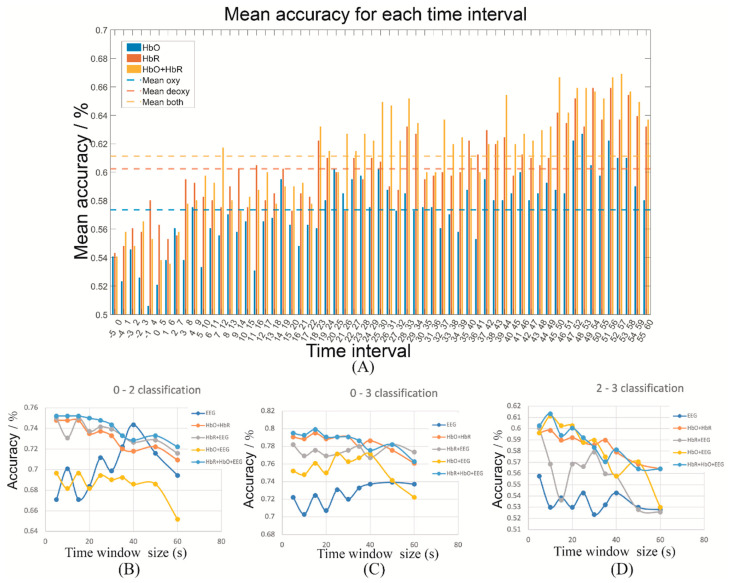
Time window analysis (**A**) Time interval analysis for fNIRS features; Time window-size evaluation for EEG and fNIRS features for (**B**) 0-back vs. 2-back, (**C**) 0-back vs. 3-back, (**D**) 2-back vs. 3-back.

**Figure 4 sensors-22-07623-f004:**
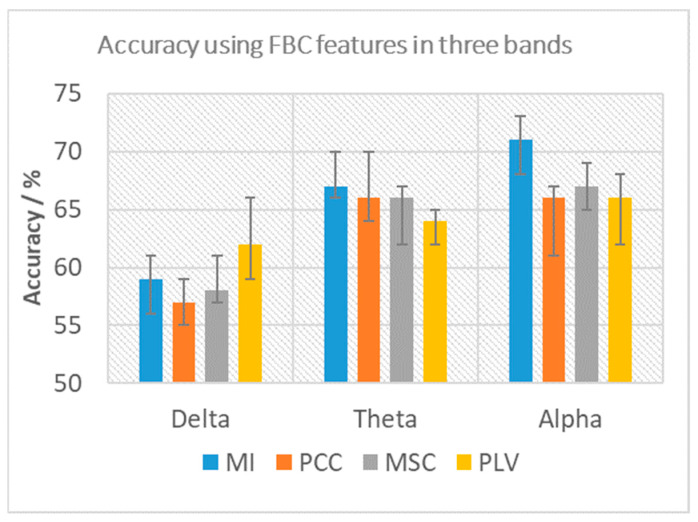
Comparison of four FBC estimations (MI, PCC, MSC and PLV) in terms of the average of the Top 10 classification accuracies along with maximum and minimum value.

**Figure 5 sensors-22-07623-f005:**
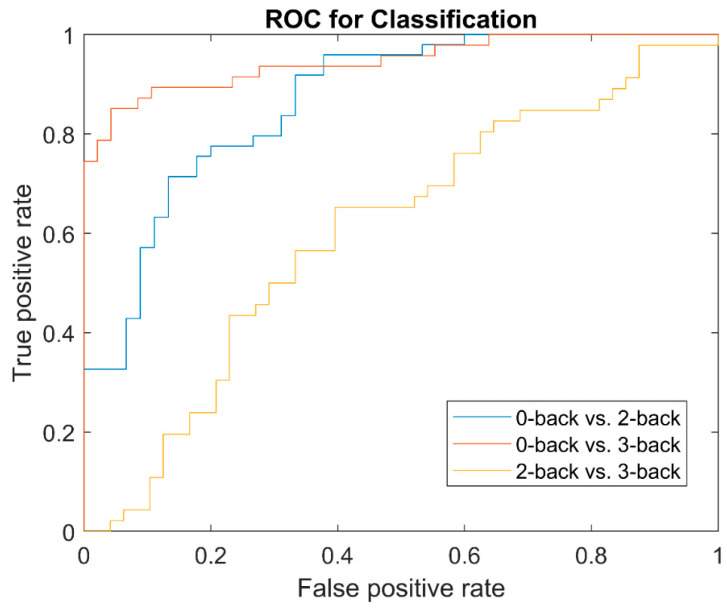
The receiver operating characteristic (ROC) curves for three binary classification tasks: 0-back vs. 2-back, 0-back vs. 3-back, and 2-back vs. 3-back.

**Figure 6 sensors-22-07623-f006:**
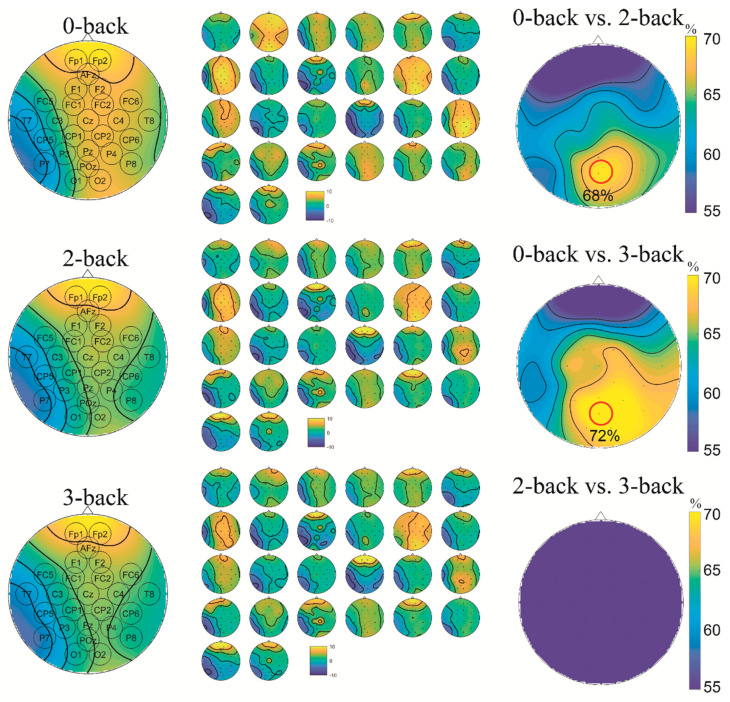
Topographic map of the EEG alpha-band PSD. Left: average; Middle: each participant; Right: Accuracy using each-channel PSD as the input. The area that provides the highest accuracy is highlighted.

**Figure 7 sensors-22-07623-f007:**
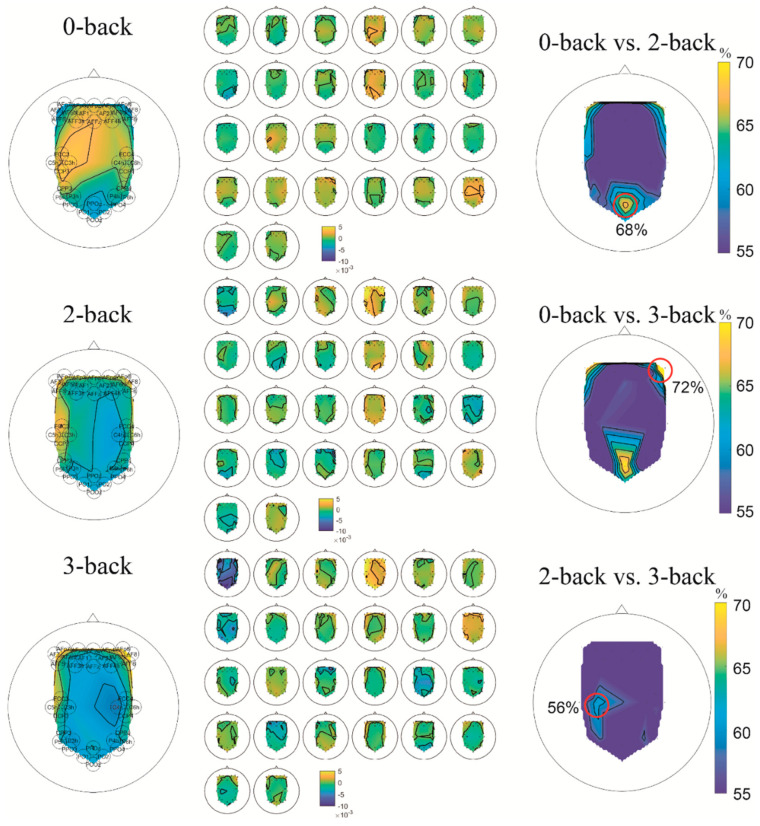
Topographic map of the fNIRS HbR features. Left: average; Middle: each participant; Right: Accuracy using each-channel HbR feature as the input. The area that provided the highest accuracy is highlighted.

**Figure 8 sensors-22-07623-f008:**
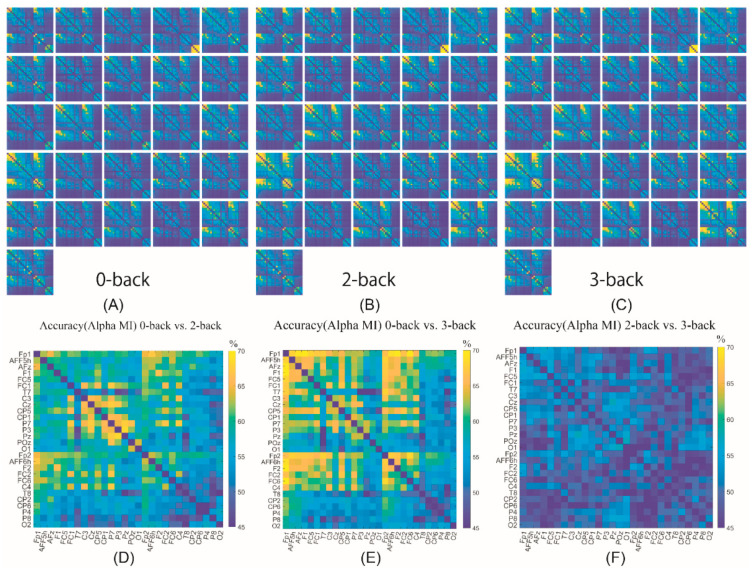
The heat map of MI FBC features and the accuracy results. (**A**–**C**) shows the MI value for each participant in 0-back, 2-back, and 3-back task. (**D**–**F**) represents the classification accuracy for 0-back vs. 2-back, 0-back vs.3-back and 2-back vs. 3-back, respectively, using each pair of EEG channels as the input where the FBC value was estimated by MI.

**Table 1 sensors-22-07623-t001:** Comparison of four neuroimaging techniques.

Specification	fNIRS	EEG/MEG	fMRI	PET
**Spatial resolution**	2–3 cm	5–9 cm	0.3 mm voxels	4 mm
**Penetration depth**	Brain cortex	Brain cortex for EEG/deep structures for MEG	Whole head	Whole head
**Temporal sampling rates**	≤10 Hz	>1000 Hz	1–3 Hz	<0.1 Hz
**Range of possible tasks**	Enormous	Limited	Limited	Limited
**Robustness to motion**	Very good	Limited	Limited	Limited
**Range of possible participants**	Everyone	Everyone	Limited, can be challenging for children/patients	Limited
**Sounds**	Silent	Silent	Very noisy	Silent
**Portability**	Yes, for portable systems	Yes, for portable EEG systems	None	None
**Cost**	Low	Low for EEG; high for MEG	High	High

**Table 2 sensors-22-07623-t002:** SVM classification accuracy of 0-back vs. 2-back using different features.

		EEG		fNIRS		EEG + fNIRS
		PSD	FBC	HbO	HbR	
**0-back vs. ** **2-back**	Delta	66%	67%	62%	68%	72%
	Theta	68%	73%			75%
	Alpha	70%	74%			**77%**

**Table 3 sensors-22-07623-t003:** SVM classification accuracy of 0-back vs. 3-back using different features.

		EEG		fNIRS		EEG + fNIRS
		PSD	FBC	HBO	HBR	
**0-back vs. ** **3-back**	Delta	65%	63%	62%	72%	74%
	Theta	69%	72%			75%
	Alpha	71%	77%			**83%**

**Table 4 sensors-22-07623-t004:** SVM classification accuracy of 2-back vs. 3-back using different features.

		EEG		fNIRS		EEG + fNIRS
		PSD	FBC	HBO	HBR	
**2-back vs. ** **3-back**	Delta	52%	60%	60%	61%	57%
	Theta	56%	61%			58%
	Alpha	55%	**62%**			59%

**Table 5 sensors-22-07623-t005:** Performance of classification for 3 binary classification tasks.

Alpha Hybrid Features	Accuracy	Specificity	Sensitivity	AUC
0-back vs. 2-back	77%	79%	76%	0.8332
0-back vs. 3-back	83%	84%	80%	0.9501
2-back vs. 3-back	59%	57%	63%	0.6721

**Table 6 sensors-22-07623-t006:** A comparative analysis of the previous research and the proposed work.

Reference	Study Setting	Classifier	Accuracy
Liu et al. [28]	0-, 1-, 2- N-back	LDA	64.4% (EEG) 55.6% (fNIRS) 68.1% (EEG+fNIRS)
Aghajani et al. [10]	0-, 1-, 2-, 3- N-back	SVM	85.9% (EEG) 74.8% (fNIRS) 90.9% (EEG+fNIRS)
Nguyen et al. [38]	Simulated driving system	FLDA	73.7% (EEG) 70.5% (fNIRS) 79.2% (EEG+fNIRS)
Saadati et al. [29]	N-back DSR Word generation LHand vs. RHand	DNN, SVM	67.0% (EEG-DNN) 80.0% (fNIRS-DNN) 87.0% (EEG+fNIRS-DNN) 82% (EEG+fNIRS-SVM)
Chu et al. [39]	Mental workload	SVM, RF, DT	55.4% (EEG-RF) 69.2% (fNIRS-RF) 78.3% (EEG+fNIRS-RF)
Proposed study	0-, 2-, 3-back	SVM	77% (0-back vs. 2-back) 83% (0-back vs. 3-back) 59% (2-back vs. 3-back)

Abbreviations: LDA—Linear discriminant analysis; SVM—Support Vector Machine; FLDA—Fisher Linear Discriminant Analysis; DNN—Deep Neural Network; RF—Random Forest; DT—Decision Tree; DSR—discrimination/selection response task.

## Data Availability

The data presented in this study are available on request from the corresponding author. The data are not publicly available due to privacy.

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
