# Peer review of "EEG/fNIRS Based Workload Classification Using Functional Brain Connectivity and Machine Learning"

_sensors, 2022, doi:10.3390/s22197623_

Round 1

Reviewer 1 Report

This study aimed to propose a machine learning architecture for mental workload classification using EEG//fNIRS. I have the following suggestions.

1.     What is the novelty of this study although several ML architectures for mental workload classification using EEG//fNIRS signal have been proposed earlier?

2.     Please write down the contribution of the study at the end part of the Introduction section in bulleted form.

3.     Authors should introduce the mental workload classification using EEG//fNIRS, such as disease prediction. Machine-learning approaches are utilized for stroke prediction in article, healthsos: real-time health monitoring system for stroke prognostics; and in article, quantitative evaluation of task-induced neurological outcome after stroke.

4.     EEG is highly sensitive to the powerline, muscular and cardiac artifacts. In EEG data preprocessing, authors need to mention how you handle AC power, EOG, and EMG artifacts in EEG signals. Do the authors think that their proposed method is robust to such kinds of artifacts?

5.     Authors need to mention the model parameters or hyperparameters of proposed ML model.

6.     It is recommendated to use explainable ML techniques (i.e. SHAP, LIME) to visualize the feature importance in ML models.

7.     Authors should provide statistical results of EEG//fNIRS parameters of the mental workload classes.

8.     Authors should present the training and validation accuracy and error graphs of the proposed model.

9.     Authors should discuss the case studies of EEG applications, such as sleep stages, driving workload , brain stimulation for different neurological workloads in article, quantifying physiological biomarkers of a microwave brain stimulation device; in article, quantitative evaluation of eeg-biomarkers for prediction of sleep stages; and in article, driving-induced neurological biomarkers in an advanced driver-assistance system.

10.  How did the authors deal with dataset class imbalance challenges in classification?

11.  Authors should report more performance measures of their model, such as, accuracy, sensitivity, specificity, precision.

12.  Both training and testing ROC curves need to be shown for each class. What ML model validation method authors used?

13.  The discussion section needs to be improved. Authors must make discussion on the advantages and drawbacks of their proposed method with other recent studies adding a table in a discussion section.

Reviewer 2 Report

A method relying on hybrid EEG-fNIRS features is proposed to deal with multi-level mental workload classification, supported by machine learning classifiers. To be more specific, the univariate PSD and four bivariate FBC features are extracted from EEG recording in three frequency bands. With the assistance of HbO and HbR indicators from fNIRS, the fused features improve the classification performance. Besides, the topographic and heat-map visualisation indicate the distinct regions for EEG and fNIRS to represent the difference among 0-back, 2-back 348 and 3-back.

As a study, the result are provided well but i am unable to reconcile whether the data was acquired for mental workload? whether the hemodynamics and EEG features were able to distinguish such load?

- It is not clear, what is the purpose of Table 1. Is it necessary to provide this comparison in context to the current study? to me, it is not a review paper. Clarification is needed. 

- In case of EEG, the volume conduction effect will come while doing the PCC.  It will not be suitable for the connectivity comparison. 

- How this range of filter between 0 and 0.04 Hz was selected? 

- Whether the data used in this work was designed to assess the mental workload? whether the reaction times were assessed? Were any behavioral results reported?

- "On one hand, EEG needs a longer data length to suggest the difference between different-level workloads" Please check whether this sentence is correct.

- The results were not compared with the existing studies. 

Round 2

Reviewer 1 Report

Thank you for addressing the comments.

1.  Authors should add more details in the captions of the figures.

2.   Figure qualities need to improve in terms of color contrast, figure text, Figure size, etc.

Author Response

The referee’ time for reviewing this paper and valuable comments are highly appreciated. All of figures and captions have been re-checked and  improved. The changes in the revised manuscript have been highlighted with the blue foreground for the Editor and referees’ convenience.  

Reviewer 2 Report

The authors have sufficiently responded well to the raised comments. it can be accepted in its current form. 

Author Response

Thank the reviewer for the positive comments.